# The Structure, Functions and Potential Medicinal Effects of Chlorophylls Derived from Microalgae

**DOI:** 10.3390/md22020065

**Published:** 2024-01-27

**Authors:** Danni Sun, Songlin Wu, Xiaohui Li, Baosheng Ge, Chengxu Zhou, Xiaojun Yan, Roger Ruan, Pengfei Cheng

**Affiliations:** 1College of Food and Pharmaceutical Sciences, Ningbo University, Ningbo 315211, China; sdnni0802@163.com (D.S.); wusonglin516@163.com (S.W.); lixiaohui@nbu.edu.cn (X.L.); zhouchengxu@nbu.edu.cn (C.Z.); 2State Key Laboratory of Heavy Oil Processing and Center for Bioengineering and Biotechnology, China University of Petroleum (East China), Qingdao 266580, China; gebaosheng@upc.edu.cn; 3Key Laboratory of Marine Biotechnology of Zhejiang Province, School of Marine Sciences, Ningbo University, Ningbo 315211, China; yanxiaojun@nbu.edu.cn; 4Center for Biorefining, Department of Bioproducts and Biosystems Engineering, University of Minnesota-Twin Cities, Saint Paul, MN 55108, USA

**Keywords:** Chlorophyll, structural function, biological activity, medicinal effects

## Abstract

Microalgae are considered to be natural producers of bioactive pigments, with the production of pigments from microalgae being a sustainable and economical strategy that promises to alleviate growing demand. Chlorophyll, as the main pigment of photosynthesis, has been widely studied, but its medicinal applications as an antioxidant, antibacterial, and antitumor reagent are still poorly understood. Chlorophyll is the most important pigment in plants and algae, which not only provides food for organisms throughout the biosphere, but also plays an important role in a variety of human and man-made applications. The biological activity of chlorophyll is closely related to its chemical structure; its specific structure offers the possibility for its medicinal applications. This paper reviews the structural and functional roles of microalgal chlorophylls, commonly used extraction methods, and recent advances in medicine, to provide a theoretical basis for the standardization and commercial production and application of chlorophylls.

## 1. Introduction

Synthetic chemicals, as a double-edged sword, not only improve our lives but also have an impact on the ecological environment and human health [1]. Encouragingly, the application of marine natural active ingredients in health care, disease prevention and treatment has also achieved promising outcomes [2]. This provides the possibility of replacing some synthetic products with natural bioactive ingredients, and promotes the development and industrialized production of natural, high-value compounds by manufacturers in the pharmaceutical and nutraceutical industries.

Microalgae are one of the oldest photosynthetic organisms in nature. Approximately 200,000 to 800,000 species of microalgae have been discovered in various environments worldwide [3]. Microalgae can respond to environmental changes by metabolically altering the characteristics of active substances within their cells. Therefore, altering culture conditions and adjusting the nutrient composition can increase the production of certain active substances in microalgae [4]. When bioactive chemicals are secreted into the medium by microalgae, exometabolism will occur. The bioactive metabolites of microalgae have a wide range of applications in the medical, pharmaceutical, cosmetic, and food industries [5].

Microalgal pigments, like chlorophylls (Chl) and carotenoids, are key marine bioactive substances with diverse health and industrial applications. It is worth noting that although there are differences in chlorophyll composition and quantity between higher plants and microalgae, their structures and functions are similar. Currently, most of the pigments on the market are chemically synthesized and are usually considered not entirely safe, or even partially toxic, due to their molecular structure or organic reactions that occur during processing. Compared with synthetic pigments, natural pigments are generally no toxic to the human body and can even be used as nutritional enhancers with a range of biological activities. Among the numerous alternatives to synthetic pigments, microorganisms, as one of the rich sources of active substances, are gradually being recognized for their environmentally friendly, low-cost, and high-yield advantages, with demand for their commercial use growing steadily. Although some plant-derived pigments have been commercialized, it has been found that the activity of pigments in microalgae is several times compared to higher plants [6]. Pigments are an important class of compounds in microalgae with powerful biological activity [7]. Most of the current data on the identification of bioactive compounds in microalgae have been focused on carotenoids [8], while relatively little research has been done on chlorophylls which constitute a significant portion of microalgae pigments [9].

Chlorophylls are a class of green pigments found in photosynthesizing organisms, including higher plants and certain types of microorganisms; they belong to the porphyrin group of compounds, including chlorophylls a, b, c, d, f and various derivatives. Chlorophyll is one of the most important ingredients in many products, and is used in the production of pharmaceutical products, in addition to being a natural food pigment [10]. Chlorophyll-a is the main photosynthetic pigment, and is abundant in cyanobacteria and red microalgae. It is worth noting that *Spirulina* is the largest source of chlorophyll-a, and serves as a colorant as an alternative to artificial colors. Gross et al. noted that in Brazil, approximately 0.06 mg/g of chlorophylls in spinach is used as a natural green pigment, while *Spirulina* biomass contains 10 mg/g of chlorophylls [11,12]. Chlorophyll and its derivatives have health-promoting properties, in addition to being used as colorants in food and pharmaceuticals. Sun et al. concluded that chlorophylls have a promising role in the prevention of nerve damage, atherosclerosis, cardiovascular diseases, metabolic syndrome, diabetes, ophthalmic diseases, viral diseases, chronic inflammatory diseases, and tumors [13].

Despite the wide range of promising applications, the commercialization of microalgal chlorophyll has been limited by factors such as its productivity and extraction efficiency [14]. Based on this, this paper reviews the potential of chlorophylls produced by microalgae as a future source of valuable macromolecules and their extraction methods. In specifically focuses on their potential applications in health, particularly in the field of medicine, and points out the limitations and prospects for the development in industries, to provide a theoretical basis for the standardization and commercial production of microalgal chlorophyll applications.

## 2. Structure and Functional Role of Chlorophylls

### 2.1. Structure of Chlorophylls

Most photosynthetic organisms, from cyanobacteria to higher plants, contain chlorophyll, differing only in the amount and type of chlorophyll contained. There are five species of Chl in algae and higher plants: a, b, c, d and f (Figure 1). Chlorophyll-b differs from chlorophyll-a by the presence of a formyl group instead of a methyl group at the C7 position. Chlorophyll-c does not have any reduced pyrrole ring and is therefore closer to the porphyrin ring, and is also known as chlorophyllide due to the presence of acrylic side chains instead of long phytol chains. The difference between chlorophyll-d and chlorophyll-a is that there is a formyl group at the C3 position instead of a vinyl group, while chlorophyll-f has only recently been isolated.

Most of the various mutants are caused by genetic mutations in the process of chlorophyll metabolism, which involves 27 genes and 15 enzymes [15]. The synthesis of chlorophyll is initiated by glutamic acid (Glu), which is catalyzed by Glu-tRNA Synthetase (GluRS). The resulting Glutamyl-tRNAGlu from this reaction is generally considered to be the starting point of chlorophyll synthesis. Catalyzed by Glutamyl-tRNA reductase (GluTR), Glutamy-tRNAGlu is reduced to Glutamate-1-semialdehyde (GSA) and intact tRNA is released. GSA is catalyzed by Glutamate-1-semialdehyde aminotransferase (GSAT) to form δ-aminolevulinic acid (ALA). Subsequently, ALA is catalytically condensed by porphobilinogen synthase (PBGS) or ALA-dehydratase (ALAD) to form monoporphyrin porphobilinogen (PBG). PBG is catalyzed by porphobilinogen deaminase (PBGD) or Hydroxymethylbilane synthase (HMBS) to form a linear hydroxymethylbilane (Hmb). Uroporphyrinogen III synthase (UROS) catalyzes the intramolecular rearrangement of linear Hmb to form uroporphyrinogen III (Urogen III), the common precursor of all natural tetrapyrrole compounds. Under the catalysis of Uroporphyrinogen decarboxylase (UROD), four acetate groups in the side chain of Urogen III were decarboxylated to synthesize coproporphyrinogen III (Coprogen III Coprogen III was decarboxylated and catalyzed by Protoporphyrinogen IX oxidase (PPO) to form protoporphyrin IX (Proto IX). Proto IX is a watershed in the tetrapyrrole biosynthesis pathway, where one is the iron branch, involved in the synthesis of Heme and Phytochrome catalyzed by ferrous chelatase (FeCH). The other is the magnesium branch for the synthesis of Chl, which is catalyzed by the involvement of Magnesium chelatase (MgCh) to give Mg-protoporphyrin IX (MgP IX). The magnesium chelation reaction is a hallmark of the entry of Proto IX into the chlorophyll synthesis pathway, while MgCh is also one of the key enzymes for chlorophyll biosynthesis. MgP IX was methylated to Mg-protoporphyrin IX monomethyl ester (MgPME) catalyzed by S-adenosylmethionone Mg-protoporphyrin IX monomethylester transferase (ChlM). MgPME is cyclized to Divinyl protochlorophyllide catalyzed by Mg-protoporphyrinogen IX monomethylester cyclase (MgPEC). Divinyl protochlorophyllide is catalyzed by 3,8-Divinyl protochlorophyllide an 8-vinyl reductase (DVR), in which the vinyl group at the C-8 position is converted to an ethyl group to produce Protochlorophyllide (Pchlide). Pchlide requires protochlorophyllide oxidoreductase (POR) to catalyze its reduction to chlorophyllide (Chlide) under light conditions. The methyl group at position C-7 of Chlide-a is catalyzed by chlorophyllide *a* oxygenase (CAO) to form Chlide-b. Finally, Chlide-a or Chlide-b is catalyzed by chlorophyll synthase (CHLG) to form Chl-a or Chl-b. The anabolic process of chlorophylls is shown in Figure 2.

Chlorophyll molecule consists mainly of two parts, the porphyrin ring containing magnesium atoms and chlorophyll alcohol. According to the number of vinyl side chains on the porphyrin ring, chlorophylls of aerobic photosynthetic organisms can be divided into two groups. Among them, chlorophylls containing one vinyl group in each of the C-3 and C-8 positions of the porphyrin ring are called divinyl chlorophyll (DV-Chl), and chlorophylls containing only one vinyl group in the C-3 position of the porphyrin ring and an ethyl group in the C-8 position are called monovinyl chlorophyll (MV-Chl). Almost all aerobic photosynthetic organisms contain only MV-Chl under normal growth conditions, but Chlorella (Prochlorococcus marinus) contains only DV-Chl. During chlorophyll biosynthesis, the conversion of the 8-vinyl group of DV-type intermediates to ethyl groups is catalyzed by the enzyme 8-vinyl reductase, resulting in the formation of normal MV-Chl [16]. Chlorophyll synthesis in microalgae is a complex process, which is regulated by external environmental conditions, such as light and rare earth elements; in addition, its intrinsic gene regulation is also very important. To date, all 15 steps in the biosynthetic pathway from L- glutamyl-tRNA to chlorophyll-a and chlorophyll-b have been identified [17].

### 2.2. Functional Role of Chlorophylls

Chlorophyll is an important pigment found in green plants and algae that is involved in photosynthesis, and plays an important role in the energy capture and energy transfer of photosynthesis. It captures light energy and uses the photolysis of water molecules to replenish the reducing capacity of the cells, which is required for carbon assimilation in the subsequent steps of photosynthesis. The energy absorbed by chlorophyll is efficiently utilized in photochemical reactions during the preparatory phase of photosynthesis, and to replenish the reducing capacity and ATP for the cells. It is subsequently utilized in biochemical reactions in the next phase of photosynthesis. All chlorophyll molecules absorb electromagnetic energy at certain wavelengths, but only a few molecules can convert this energy into chemical energy [18].

Chlorophyll acts as the central pigment of photosynthesis and drives the process of photosynthesis. During photosynthesis, chlorophyll accepts the energy and transfers it to other pigments, such as carotenoids and lutein. These pigments then pass the energy to reaction centers, which are crucial in converting carbon dioxide and water into organic matter through photosynthesis. Chlorophyll can also participate in the synthesis of other pigments. For example, in the synthesis of carotenoids, chlorophyll is involved as an intermediate in the process of their synthesis [19]. However, the mechanism of chlorophyll involvement in the synthesis of other pigments may vary for different species of microalgae. In *Spirulina*, there are two main pathways in which chlorophyll is involved in the synthesis of carotenoids: direct and indirect. The direct pathway refers to the fact that chlorophyll participates in its synthesis by binding directly to carotenoids; the porphyrin ring in the chlorophyll molecule can combine with a hydrocarbon chain in the carotenoid molecule to form the chlorophyll-carotenoid complex [20]. This complex can be further converted into other types of carotenoids, such as β-carotene and lutein. The indirect pathway refers to the synthesis of other intermediate products, such as fatty acids and cholesterol by chlorophyll, which are then involved in the synthesis of carotenoids. The chlorophyll a in *Spirulina maxima* can be involved in the regulation of lipogenesis pathway enzyme, and inhibit lipid accumulation [21]. The function role of chlorophylls is shown in Figure 3.

Chlorophyll plays a crucial role in providing food for organisms across the biosphere through its key role in photosynthesis. With an understanding of the process of light reactions, the organization of different chlorophyll molecules, and the coordinated functions of other molecules, scientists and engineers have devised several strategies to develop chlorophyll molecules for a wide variety of applications.

The interaction of chlorophyll with the visible spectrum of electromagnetic radiation makes it useful in spectroscopic studies and applications [22]. Wang et al. designed a chlorophyll-based fluorescent biosensor for detecting the viability of microalgae cells in ship ballast water [23]. Sanders et al. developed a tissue biosensor by immobilizing photosynthesizing *Chlorella vulgaris* and cyanobacteria *Nostoc* commune on glass fiber filter discs [24]. Chlorophyll has also been observed to possess other important activities that increase the suitability of the molecule for different applications.

In summary, chlorophyll is a versatile molecule with a range of activities extending beyond its vital role in sustaining Earth’s biosphere. Moreover, the function application of chlorophyll is also in multiple fields of human activities.

## 3. Common Extraction Methods for Chlorophylls

Compared to higher plants and microalgae, the time required for organic solvent extraction of chlorophyll is different. The extraction and release time of chlorophyll is not only related to the solvent, but also to the plant material. The chlorophyll content of microalgae is relatively high, but the extraction and release time of microalgae chlorophyll is also relatively long. The most commonly used extraction methods for chlorophyll include solvent extraction, supercritical fluid extraction, and the high-voltage pulsed electric field method [25]. However, each of these techniques has its shortcomings. Table 1 summarizes the commonly used extraction methods for chlorophylls. To realize the broad and effective application of natural pigments, the improvement of their extraction methods is a crucial aspect of the extraction process.

### 3.1. Extraction Method

Solvent extraction is one of the easiest and most used methods for pigment extraction and has been applied industrially. The Chlorophyll molecule contains a porphyrin ring “head” and chlorophyll alcohol “tail”, so it is insoluble in water, but soluble in organic solvents. So, generally used organic solvents include ethanol or acetone for extraction. This conventional extraction technique has some drawbacks, such as the use of large amounts of toxic organic solvents, long extraction times, low selectivity, low extraction rates, and exposure of the extract to excessive heat, light, and oxygen. Researchers determined the chlorophyll content of the extracts obtained through DMSO and 96% ethanol leaching, by placing them under natural light exposure for 20, 40, and 60 min, respectively [27]. They found that the chlorophyll solution extracted by 96% ethanol should not be exposed to the light for more than 20 min, and that it should be stored under strict protection from light during the extraction process.

Ultrasound-assisted extraction technology has higher extraction efficiency than solvent extraction alone, and is a low-temperature extraction (40–60 °C), requiring 20–40 min of extraction time, and is a high extraction efficiency (more than 50% increase) auxiliary extraction method. The ultrasonic waves generated by this method can shuttle and vibrate in the liquid solvent, producing ultrasonic “blasting” and promoting the output of natural pigments [29]. However, the noise pollution of ultrasound has an impact on its industrialization [30].

Pressurized liquid extraction (PLE) is a method that has recently been used to extract biologically active components [32]. By using conventional solvents at high temperatures and pressures, this method is an environmentally clean extraction method. PLE has the potential to be a potent tool in various industries due to its automation, reduced solvent usage, shorter processing time, and the ability to maintain samples in an oxygen-free and light-free environment. Kwang et al. extracted chlorophyll and carotenoids from green microalga *Chlorella vulgaris* by (1) leaching (MAC), (2) ultrasound-assisted extraction (UAE), and (3) pressurized liquid extraction (PLE). These three extraction processes were compared, and PLE proved to have higher extraction efficiency than the MAC or UAE methods [33].

Compared to other extraction methods, supercritical fluid extraction (SCF) is a thriving new technology that processes physical properties in three states: gas, liquid, and solid. In the context of natural pigment extraction, the physical and chemical properties of supercritical fluids (such as density, viscosity, and diffusion rate) are between liquid and gas [41]. This method offers several advantages, including a low extraction temperature, a high fast mass transfer speed, and a high extraction speed. The most commonly used supercritical fluid extraction material is CO_2_. In systems where CO_2_ is used as a supercritical fluid extraction reagent, the introduction of substances like hydrocarbons (such as ethanol or methanol) and natural oils can enhance CO_2_ affinity for various solutes, thereby improving the extraction efficiency of natural pigments [37]. SCF, as a green analysis method for extracting high-value bioactive compounds from complex matrices [42], has broad application prospects for different bioactive compounds [43]. Fabrowska et al. used SCF to extract pigments such as carotenoids and chlorophylls from Arctic brown algae, and found in subsequent studies that the extract exhibited significant bactericidal, fungicidal, and immunostimulatory activities [44,45].

### 3.2. Other Methods

Enzyme-assisted extraction involves the selection of suitable enzymes to break down the cell walls under relatively mild reaction conditions, allowing the target active ingredients to flow more readily into the extraction medium [38]. Solvent extraction, and other methods, can be used as separate extraction methods to extract chlorophyll, however, these methods need to be heated to remove the organic solvent after the extraction of the pigment, and the heat required for heating at this time is likely to affect the stability of the chlorophyll. In order to obtain highly stable natural chlorophylls, commercial enzymes can be used for protein lysis, followed by ultrafiltration to separate proteins from pigments.

High voltage pulsed electric field (PEF) method is an extraction method with short processing time, low processing temperature, long shelf life, and high extraction rate. For natural pigments, the extraction rate largely depends on the conditions of the biological material being processed, particularly the extent to which it has been broken down or disrupted. Traditional cell wall breaking methods include physical vibration, impact method, chemical decomposition method, and biological wall breaking method [39]. PEF is a novel wall-breaking technique in which complex biological samples are placed between two electrodes exposed to a high-intensity electric field. A high voltage can then be applied in the form of repetitive pulses, typically ranging from a few nanoseconds to a few milliseconds in duration. La et al. demonstrated that electroporation can improve the permeability of cell membranes and promote the release of intracellular substances [46]. Pataro et al. showed that high-pressure pulse treatment effectively increased the yield of extracted carotenoids and their antioxidant capacity, and no isomerization or degradation of carotenoids occurred during the extraction process [47]. These studies suggest that PEF is a more warming and efficient pre-treatment for cell breakdown in wet plant tissues than typical extraction techniques. But there are fewer research results available in the current literature, on the extraction of chlorophyll using high-voltage pulsed electric fields.

In recent years, various scales of equipment and industrial prototypes for PEF have been developed [48]. But electric field assistance is still a relatively novel and advanced extraction technology. The reason for this is that the method requires high voltage pulses with sufficient peak power. As a result, PEF, while effective in laboratory settings, has yet to achieve widespread industrial-scale application [40].

## 4. Medical Application of Chlorophylls

The biological activity of chlorophyll is closely related to its chemical structure. For example, substitution of the central magnesium ion by other metals (zinc or copper) alters the ability to bind proteins and affects the stability and kinetics of chlorophyll during organization transport. On the other hand, the presence of phyllotactic chains increases the hydrophobicity of the molecules and thus improves the aggregation of molecules. Additionally, rearrangements at peripheral positions of different chlorophylls, or homocycles of oxidized chlorophylls, can change the nature of the p-electron system. This alters the absorption properties and the corresponding specific bioactive properties. This enables chlorophyll and its derivatives to function as antioxidants, antimicrobials, cancer prevention agents, and anti-mutagenic activity (Figure 4).

### 4.1. Antioxidant

Compared to research on carotenoids, there is limited understanding of the yield, absorption and mass-transport processes, metabolic pathways, and precise oxidation mechanisms of chlorophyll metabolites. The structure and conformation of chlorophyll affects its antioxidant activity and therefore different studies have addressed this issue. The first step is to clarify the differences between the Chl a and Chl b lines, as the chlorophyll results of the a and b lines are contradictory [49]. For example, some reports have found that b-series chlorophyll compounds have higher antioxidant activity than a-series chlorophyll compounds, indicating that the important role of aldehyde groups at the C7 in antioxidant capacity which is still unclear [50,51]. On the contrary, in similar experiments, Hsiao et al. [52] showed that the free radical quenching effect of chlorophyll-a was three times that of chlorophyll-b. This is consistent with the previously reported results of iron triacetate-induced singlet oxygen lipid peroxidation experiments [53] or CUPRAC experiments [54].

Metal-free chlorophyll derivatives (chlorophyllotoxins and sulfur-repellent agents) were significantly less resistant to free radicals than metal-containing chlorophyll derivatives, where the presence of copper was more favorable than the presence of zinc or magnesium. It is speculated that the presence of metals can increase the electron density of the skeleton center, thereby enhancing the ability of conjugated porphyrin chains to donate electrons [55,56]. However, it was found that the antioxidant capacity of chlorophyll is higher than that of metallochlorophylls. It is speculated that non-metallic chlorophylls may exert their antioxidant capacity through their intrinsic ion-chelating ability and porphyrin stabilization of reactive oxygen species [57].

The antioxidant properties of chlorophylls have been investigated with the most common chlorophyll derivatives. Sodium copper chlorophyllin (SCC), due to the characteristics of no cyto/genotoxic properties toward human blood cells, can be produced from natural chlorophyll by solvent saponification reactions and copper incorporation (Figure 5). In addition to its attractive coloring properties, this group of chlorophyll derivatives consistently showed the highest antioxidant activity; and it was three to five times higher compared to chlorophyll derivatives not bound to copper [55]. Consistent results were obtained through in vitro analytical methods, including DPPH, ABTS, and the β-carotene test. These methods quantitatively determined the presence of free radicals, which caused the decolorization of β-carotene during the process of fatty acid oxidation [55,58,59]. Using the comet protocol, or biomarkers such as MDA, the oxidative protective effect of copper chlorophyll was demonstrated in cultured human blood cells [60]. Sato et al. investigated the antioxidant activity of SCC salts in rat hepatocytes, and found that thiobarbituric acid completely inhibited the production of reactive substances, and retarded lipid peroxidation. Similarly, SCC salt was able to reduce the formation of lipid hydroperoxides and avoid the activation of superoxide dismutase (SOD) against oxidative damage induced by methylene blue, ascorbic acid-Fe^2+^, and NADPH-ADP-Fe^3+^, even more effectively than other antioxidants, such as ascorbic acid, mannitol, and glutathione (GSH) [61].

The antioxidant capacity of chlorophylls is also being studied. Batista et al. utilized the biomass of Arthrospira platensis, Chlorella vulgaris, Tetraselmis suecica, and Phaeodactylum tricornutum, to make wheat crackers. The antioxidant capacity of these crackers was determined using the DPPH method. Interestingly, it was observed that as the microalgae biomass content in the cookies increased, the antioxidant capacity of the crackers also increased. This highlights the potential of microalgae-derived chlorophylls to enhance the antioxidant properties of food products [62]. Through a different approach, utilizing the peroxyl radical scavenging assay, it was discovered that the chlorophyll fraction found in Phormidium autumnale exhibited remarkably high antioxidant activity. In fact, its antioxidant capacity was estimated to be approximately 200 times greater than that of α-tocopherol, highlighting the potent antioxidant properties of this chlorophyll fraction [63]. Kummar et al. investigated the antioxidant effect of chlorophyllin against different ROS using electron spin resonance spectroscopy (ESR); he also concluded that SCC could prevent the damage caused by radiation from different sources, and that the antioxidant effect was concentration-dependent [64]. Chlorophyll and pheophytin showed lipid antioxidants in stored edible oils, and reduced free radical activity in standardized tests, with effects even similar to classical antioxidants such as carotenoids, vitamin E and vitamin C [65]. Recent studies have shown that the antioxidant activity of natural chlorophyll derivatives can significantly reduce the effects of some diseases associated with oxidative processes, such as inflammation, prevent oxidative DNA damage, and reduce the risk of cancer [63,66,67]. Chlorophyll and its derivative, pheophytin, have anti-inflammatory effects and exhibit anti-inflammatory activity against carrageenan-induced foot edema in mice and formalin-induced inflammation in rat [68]. Hsu et al. analyzed the role of chlorophyll derivatives in the protection of lymphocytes against oxidative DNA damage by H_2_O_2_, and explored whether they act as direct scavengers of free radicals or as chelators of Fe^2+^ [68]. The authors showed that chlorophyll derivatives act as antioxidants, and can prevent oxidative DNA damage and lipid peroxidation by both mechanisms.

A common experimental approach to study the free radical scavenging properties of chlorophyll in vivo, is to analyze different tissues of experimental animals (mice, rats, etc.) on a diet rich in chlorophyll (possibly copper chlorophyll). Excessive ROS production in cells induces damage to lipids, proteins, and DNA, and the results showed that chlorophyll was able to reduce general ROS levels at the in vitro level [69,70].

### 4.2. Antimicrobial

Zheng et al. studied the effects of chlorophylls on the intestinal microbiota, intestinal mucosal barrier, and hepatic fibrosis [71]. This work demonstrated that chlorophyll directly affects and balances the intestinal microbiota in mice, as evidenced by the down-regulation of the phylum Firmicutes and the up-regulation of the phylum Bacteroidetes. For the first time, results obtained from this study supported the role of chlorophyll derivatives as prebiotics, and provided evidence for the potential application of chlorophyll in regulating intestinal flora and ameliorating hepatic fibrosis [72]. In a study involving dogs, the combination of probiotics (specifically Lactobacillus fermentum) and chlorophyllin was investigated. The antimicrobial effects of chlorophyll were primarily observed as a reduction in the population of coliforms and Clostridia bacteria, while it had no discernible impact on the population of beneficial bacteria like Lactobacillus. This suggests that the inclusion of chlorophyllin in the diets of dogs may be beneficial in preventing intestinal bacterial infections without adversely affecting the growth of beneficial bacteria [73]. Ghate et al. detailed the use of chlorophyll compounds as microbial inactivators in food safety due to their photosensitizing properties [74]. In vitro studies have shown that chlorophylls are able to fight against different cultures of foodborne bacteria and fungi, and can even be used as decontaminating agents for fruits and vegetables, extending the corresponding shelf life.

### 4.3. Anticancer

The ability of chlorophylls to modulate oxidative stress, and limit the bioavailability of mutagens, makes it a good candidate for anticancer activity [75,76]. Compelling data from experimental and epidemiological studies have demonstrated that the anticancer effects of chlorophylls are mediated through a variety of mechanisms. These include modulation of carcinogenic metabolism, inhibition of DNA adduct formation, up-regulation of the detoxification system, and inhibition of cell proliferation by inducing cell-cycle blockade and/or apoptosis [77,78].

Currently, most of the in vitro studies on anticancer activity have been performed with chlorophyll directly, or the synthetic SCC [79]. The mechanism seems to be related to the formation of a complex between the tetrapyrrole and the mutagen, which promotes the elimination of the mutagen through the digestive tract [71]. These complex reduces the uptake and bioavailability of carcinogens, enhances the elimination of unmetabolized carcinogens, inhibits carcinogenic activation, and promotes antioxidant activity and induces apoptosis of cancer cells. However, studies on rats have shown that natural chlorophyll can inhibit heme induced colon cytotoxicity, colon epithelial cell proliferation, and lipid free radical formation, but SCC cannot prevent these heme induced effects [80]. At the in vivo level, different studies in animals and humans, have shown that the chlorophylls may inhibit the development or progression of cancer through a variety of mechanisms (Figure 6) [81]. In a study using the Ames test on Salmonella typhimurium, the authors found that SCC salts were effective in reducing damage caused by chemicals and complex mixtures (e.g., aflatoxin B, benzo[α]pyrene, N-methyl-N’-nitro-N-nitrosoguanidine) [82]. Chlorophyll-b has been associated with a chemopreventive effect on the side effects of cisstaplin (cDDP), a drug commonly used to treat hematological tumors. The possible antitoxic effect of chlorophyll-b on cDDP-induced DNA damage in mouse kidney, liver and peripheral blood cells, has been observed by electrophoretic detection [83]. SCC salt inhibits cancer development and progression by targeting multiple molecules and pathways involved in carcinogenic metabolism, cell cycle progression, apoptosis evasion, invasion, and angiogenesis [72]. Reddy et al. also found that SCC salts reduced the incidence of liver, stomach, and bladder tumors, and were more effective when used concurrently with benzo[α]pyrene (B-[α]-p) [84].

In addition, chlorophylls have chemopreventive effects on different tumors by targeting multiple oncogenic signaling pathways and transcription factors such as Wnt, NF-κB, TGFβ, and their downstream genes [85,86,87].

NF-κB is a convergence point for most oncogenic pathways, and typically the Wnt/β-catenin signaling pathway along with the vascular endothelial growth factor (VEGF) signaling pathway are the most prominent pathways affected by NF-κB [88]. Wnt signaling plays a central role in the regulation of cell proliferation, apoptosis, epithelial mesenchymal transition (EMT), and angiogenesis, mediated through interactions between β-catenin to members of the T cytokine/lymphocyte enhancer factor (TCF/LEF) family of transcription factors [89]. The pro-angiogenic factors VEGF and hypoxia-inducible factor-α (HIF-1α) are released during extracellular matrix (ECM) processing, triggering a network of signaling pathways that promote angiogenesis [90,91]. VEGF also regulates the pro-survival phosphatidylinositol 3-kinase/Akt signaling pathway, which is one of the key factors in NF-κB activation and apoptosis inhibition [92]. Siddavaram et al. demonstrated that dietary added chlorophyll was able to target a variety of key signaling molecules and block the Wnt/β-catenin and VEGF signaling pathways in a 7,12-dimethylbenz(a)anthracene (DMBA)-induced hamster buccal pouch (HBP) carcinoma model. Based on this, the authors suggested that chlorophyll, which inactivates NF-κB, may also have an impact on the NF-κB-driven signaling pathways that promote tumor progression [85].

Thiyagarajan et al. added chlorophyll (4 mg/kg bw) to a hamster oral tumor model and demonstrated that chlorophyll was able to inhibit the NF-κB signaling pathway (down-regulation of IKKb inhibited the NF-κB signaling pathway, prevented IκB-α phosphorylation, and decreased the expression of nuclear NF-κB). This induced endogenous apoptosis and thus inhibited the development of oral cancer [86].

Typical Smad-mediated TGFβ signaling plays a key role in the development and progression of several malignant tumors [93]. Thiyagarajan et al. demonstrated that chlorophyll inhibits N-methyl-N′ -N-nitro-N-nitrosoguanidine (MNNG)-induced anterior gastric carcinoma in vivo by modulating the TGFβ signaling pathway, including down-regulation of the expression of TGFβ RI, TGFβ RII, and Smad 2 and 4 and up-regulation of Smad 7. Nitro-N-nitrosoguanidine (MNNG)-induced antral cancer development thereby eliminated the typical TGFβ signaling pathway, which demonstrated the chemopreventive potential of chlorophyll in a rat model of antral cancer [87].

However, the most promising biological activity of chlorophyll compounds may be their role as “drugs” in photodynamic therapy (PDT), as their photophysical properties make them ideal photosensitizers. Since the development of the first photosensitizer, Photofrin1, there has been a continuous effort to enhance the structural configuration of chlorophyll through successive generations of advancements [94]. Notably, recent advances in this field involve the utilization of microencapsulation techniques aimed at reducing efflux and enhancing the overall efficacy of chlorophyll-based photosensitizers [95,96]. In addition, the photosensitizing properties of chlorophylls have been used for cancer detection by fluorescence, magnetic resonance, or nuclear imaging to obtain an early and accurate diagnosis [97,98].

### 4.4. Other Effects

The process of chlorophyll digestion in the human body has also attracted extensive research by scholars. Chlorophylls are prone to structural modifications when exposed to external factors such as pH, enzymatic action, and oxidation. To mitigate the instability of chlorophyll outside its natural environment, different microencapsulation techniques, such as spray drying, complex coalescence, and microfluidization, have been used [52,99,100]. During digestion, chlorophyll is susceptible to transformations, especially on C-132, C-173, chelate and porphyrin structures. Chlorophyll has been shown to have a similar digestive pathway to that observed in xenobiotics. The digestion process usually involves the release of compounds from food, interaction of the released compounds with gastrointestinal fluids, solubilization of water-soluble compounds in the aqueous phase, or incorporation of lipophilic compounds in mixed micelles [67]. Israel et al. extracted and microencapsulated chlorophyll using different carriers, and performed in vitro digestion. He found that the type of carrier agent can inhibit or modulate the bioaccessibility of chlorophyll, which has the potential to be an effective delivery system for health-promoting compounds [101]. Different SCC complexes have been studied in relation to skin disorders. Vasily et al. developed a formulation based on copper chlorophyll for the treatment of rosacea, a chronic skin condition. This work resulted in favorable clinical outcomes for patients in terms of reduced facial redness [102].

In addition, Choi et al. found that chlorophyll a extracted from *Spirulina* had high neuroprotective activity after optimizing conditions. So, it is proposed that chlorophyll a can prevent memory disorders and the related diseases [103]. Other benefits of chlorophyll, such as controlling the body odor of feces and urine found in elderly patients, have led to its widespread use as a nutritional supplement [104].

## 5. Conclusions and Outlook

It is predicted that the market value of algal pigments is likely to grow at a compound annual growth rate (CAGR) of 4% from 2019 to 2025. It is expected that the market value may reach US$452.4 million by 2025 [105]. However, despite the increasing use of microalgae for biopigment production as a sustainable natural source, there remain several challenges in the production technology of microalgal pigments. These challenges encompass the need for enhanced microalgal bioproduction, the need to reduce capital and operating costs (e.g., energy costs the improvement of algal pigment recovery technologies), and the challenges in scale-up production processes [6]. In addition, a lack of sufficient understanding of microalgal metabolic mechanisms is one of the main challenges to optimizing microalgal chlorophyll biosynthesis and large-scale production [106].

Microalgae have the potential to be a valuable source of chlorophyll, with its associated antioxidant, antimicrobial, and anticancer properties. However, to harness these medicinal applications fully, additional research is essential to confirm and demonstrate their effectiveness. Only then, can the potential market for microalgal chlorophyll have a broader prospective for commercial growth.

## Figures and Tables

**Figure 1 marinedrugs-22-00065-f001:**
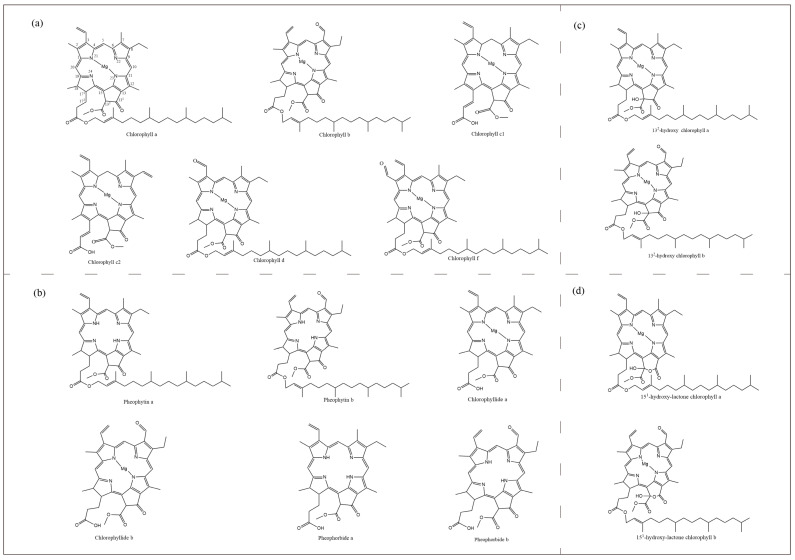
Molecular structure of chlorophylls and their derivatives. (**a**), Five species of chlorophylls in algae and higher plants. (**b**), Pheophytin a, Pheophytin b, Chlorophyllide a, Chlorophyllide b, Pheophorbide a, Pheophorbide b. (**c**), 13^2^-hydroxy chlorophyll a, 13^2^-hydroxy chlorophyll b. (**d**), 15^1^-hydroxy-lactone chlorophyll a, 15^1^-hydroxy-lactone chlorophyll b.

**Figure 2 marinedrugs-22-00065-f002:**
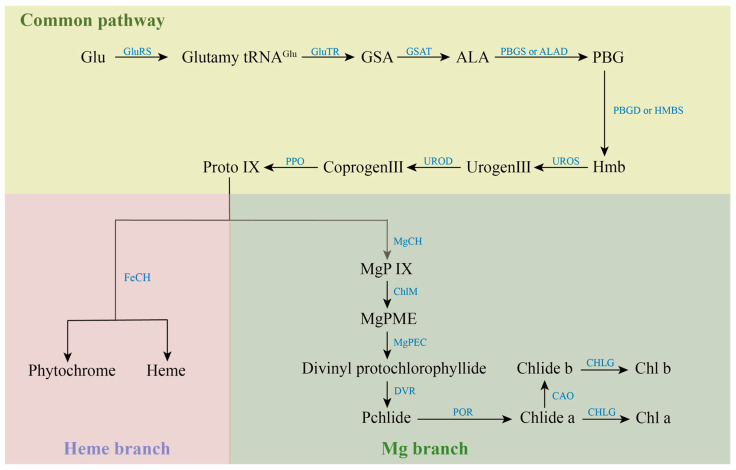
Anabolic processes of chlorophylls.

**Figure 3 marinedrugs-22-00065-f003:**
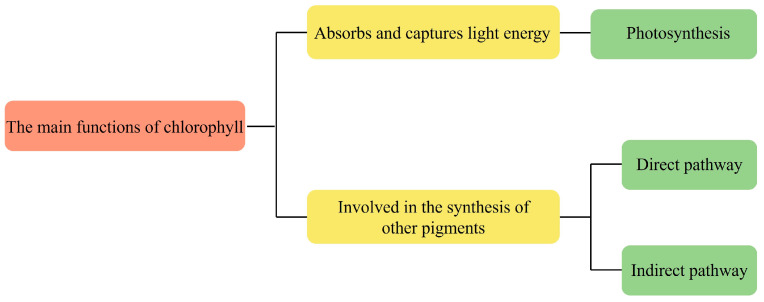
The function role of chlorophylls.

**Figure 4 marinedrugs-22-00065-f004:**
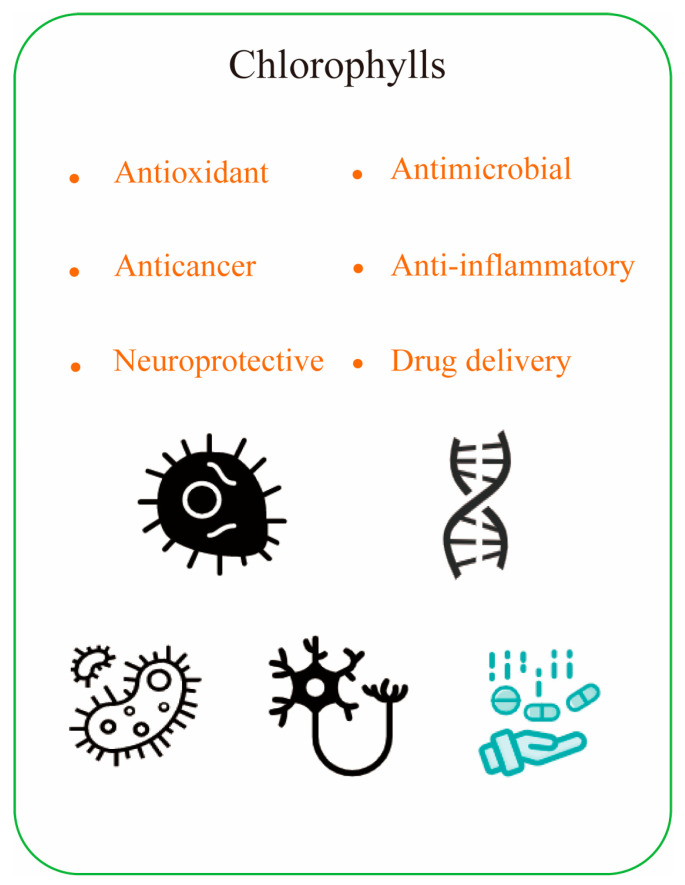
The health benefits of chlorophyll compounds.

**Figure 5 marinedrugs-22-00065-f005:**
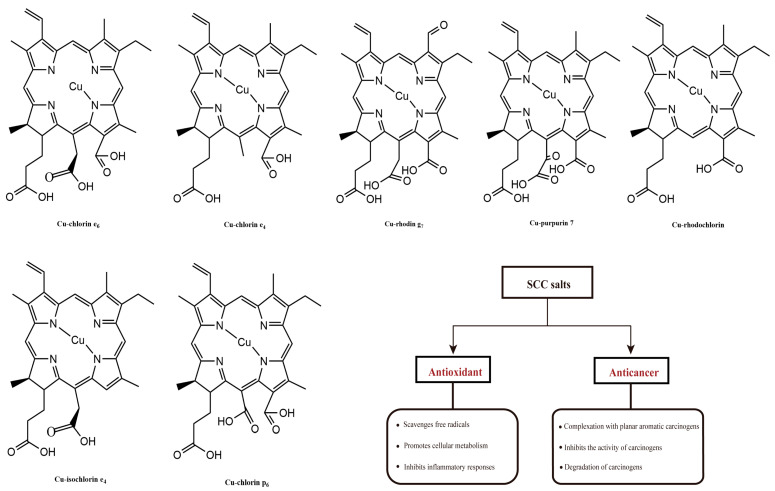
The molecular structures of SCC salts and their biological functions.

**Figure 6 marinedrugs-22-00065-f006:**
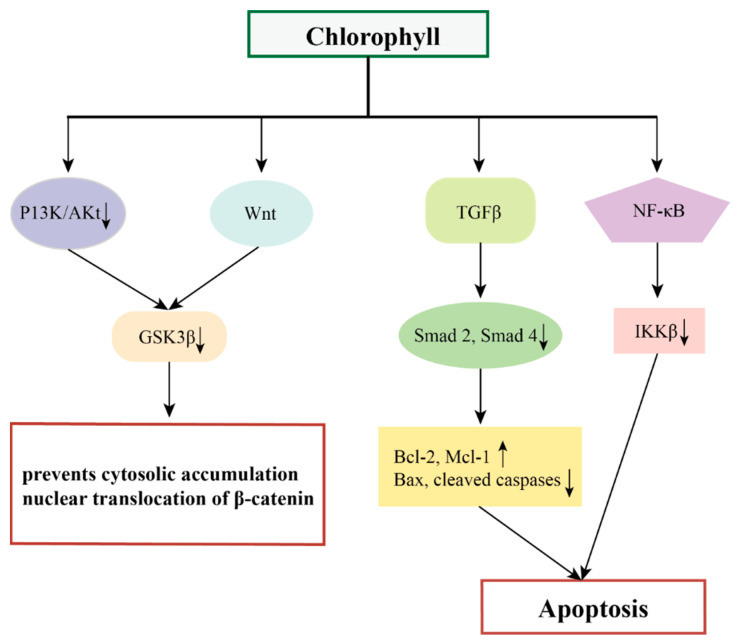
The possible mechanisms of chlorophylls in preventing cancer.

**Table 1 marinedrugs-22-00065-t001:** Summary of commonly used extraction methods for chlorophylls.

Common Extraction Methods	Name of the Specific Methods	Characteristic	Advantage		Disadvantage		References
Extraction methods	Solvent extraction	Extraction methods using organic solvents	Easy operation		Low efficiency	May be toxic	[26,27]
	Ultrasound-assisted extraction technology	Utilizes ultrasonic energy to rapidly dissolve soluble components of a substance.	High efficiency	Simple operation	noise pollution		[28,29,30]
	Pressurized liquid extraction	A high pressure solvent is utilized to dissolve the target compounds in the sample at elevated temperatures.	Environmentally friendly	Highly efficient	cumbersome operation		[31,32,33]
	Supercritical fluid extraction	Supercritical fluid extraction materials (typically CO_2_) combined with organic solvents to increase extraction efficiency	Retention of activity	High safety	High equipment costs	Strict operating conditions	[34,35,36,37]
Other methods	Enzyme-assisted extraction	Selection of the appropriate enzyme disrupts the cell wall of the cell, allowing efficient flow of the target component into the medium.	Mild reaction		Low extraction rate	Unclear conditions of action	[38]
	High-voltage pulsed electric field method	Facilitates the release of intracellular substances by altering the osmotic action of the cell membrane.	High efficiency	Long shelf life of the products obtained	Strict operating conditions for equipment		[39,40]

## Data Availability

The original data are available from the correspondent author on request.

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
