# Peer review of "The Structure, Functions and Potential Medicinal Effects of Chlorophylls Derived from Microalgae"

_marinedrugs, 2024, doi:10.3390/md22020065_

Round 1

Reviewer 1 Report (Previous Reviewer 1)

Comments and Suggestions for Authors

This is a revision of the manuscript previously submitted to Marine Drugs and reviewed by me. The authors had dealt with the issues raised in a satisfactory manner. The manuscript is suitable for publication now.

Comments on the Quality of English Language

Some language polishing is required, can be done during copyediting.

Author Response

This is a revision of the manuscript previously submitted to Marine Drugs and reviewed by me. The authors had dealt with the issues raised in a satisfactory manner. The manuscript is suitable for publication now.

Response: Thank you so much for your comments.

Reviewer 2 Report (Previous Reviewer 2)

Comments and Suggestions for Authors

Upon careful reevaluation of the revisions made by the authors in response to my previous review, I regret to inform the author that I find the issues I raised have not been adequately addressed. While the authors made some modifications in their response, I believe these changes fail to resolve the core concerns I raised, and in certain aspects, introduce new issues. The main reasons for rejecting the manuscript are as follows:

1. Contradiction: The first sentence of introduction emphasizes the potential hazards of synthetic chemicals, but later sections extensively discuss the health benefits of artificially synthesized pigments (SCC). This creates a contradiction within the manuscript.

2. Relevance of Topic and Content: While the revisions focus on structural aspects, highlighting similarities in chlorophyll structure between higher plants and microalgae, the manuscript still lacks explicit clarification regarding differences in other aspects. Specifically, the extraction section should elaborate on the distinctive characteristics of microalgae and highlight the associated advantages.

3. Concept Confusion: The distinction between natural chlorophyll and synthetic chlorophyll derivatives remains unclear. For instance, in the citation of reference 68 from lines 382-389, the material used in that reference is chlorophyllin (copper gluconate). However, in the manuscript, the author directly describes it as chlorophyll, creating confusion. This inconsistency occurs in multiple instances throughout the paper.

4. Structural and Logical Issues: The overall logic of the content still lacks substantial alteration. Additionally, the modifications made to Figure 1 appear to be minimal.

Comments on the Quality of English Language

Several lengthy sentences could be broken down for clarity. 

Author Response

Reviewer 2:

Upon careful reevaluation of the revisions made by the authors in response to my previous review, I regret to inform the author that I find the issues I raised have not been adequately addressed. While the authors made some modifications in their response, I believe these changes fail to resolve the core concerns I raised, and in certain aspects, introduce new issues. The main reasons for rejecting the manuscript are as follows:

  1. Contradiction: The first sentence of introduction emphasizes the potential hazards of synthetic chemicals, but later sections extensively discuss the health benefits of artificially synthesized pigments (SCC). This creates a contradiction within the manuscript.

Response: Thank you for your kind comments. Based on the opinions of reviewer, we have revised the first sentence of introduction. Moreover, it was reported that Sodium copper chlorophyllin (SCC), as a food colorant, had no cyto/genotoxic properties toward human blood cells [1], please refer to the section marked in blue.

Reference:

Gerić M., Gajski G., Mihaljević B., Miljanić S., Domijan A. M., Garaj-Vrhovac V., 2019. Radioprotective properties of food colorant sodium copper chlorophyllin on human peripheral blood cells in vitro [J]. Mutation Research - Genetic Toxicology and Environmental Mutagenesis, 845, 403027.

  1. Relevance of Topic and Content: While the revisions focus on structural aspects, highlighting similarities in chlorophyll structure between higher plants and microalgae, the manuscript still lacks explicit clarification regarding differences in other aspects. Specifically, the extraction section should elaborate on the distinctive characteristics of microalgae and highlight the associated advantages.

Response: Thank you for your comments. Compared to higher plants and microalgae, the time required for organic solvent extraction of chlorophyll is different. The extraction and release time of chlorophyll is not only related to the solvent, but also to the plant material. The chlorophyll content of microalgae is relatively high, but the extraction and release time of microalgae chlorophyll is also relatively long. Moreover, according to the reviewer’s suggestion, we have revised the ‘extraction section’, please refer to the revised manuscript.

  1. Concept Confusion: The distinction between natural chlorophyll and synthetic chlorophyll derivatives remains unclear. For instance, in the citation of reference 68 from lines 382-389, the material used in that reference is chlorophyllin (copper gluconate). However, in the manuscript, the author directly describes it as chlorophyll, creating confusion. This inconsistency occurs in multiple instances throughout the paper.

Response: Thank you for your comments and suggestion, we have carefully checked and revised the whole paper, and have made some corresponding modifications for the natural chlorophyll and synthetic chlorophyll derivatives, please refer to the revised manuscript.

  1. Structural and Logical Issues: The overall logic of the content still lacks substantial alteration. Additionally, the modifications made to Figure 1 appear to be minimal.

Response: Thank you for your comments and suggestions, we have revised the whole manuscript, and also revised the Figure 1, please refer to the revised manuscript.

Figure 1. Molecular structure of chlorophylls and their derivatives. (a), Five species of chlorophylls in algae and higher plants. (b), Pheophytin a, Pheophytin b, Chlorophyllide a, Chlorophyllide b, Pheophorbide a, Pheophorbide b. (c), 132-hydroxy chlorophyll a, 132-hydroxy chlorophyll b. (d), 151-hydroxy-lactone chlorophyll a, 151-hydroxy-lactone chlorophyll b.

Several lengthy sentences could be broken down for clarity. 

Response: Thank you for your comments and suggestions, we have revised the long sentences of the text, please refer to the revised manuscript.

Reviewer 3 Report (New Reviewer)

Comments and Suggestions for Authors

Add reference to Table 1

The resolution for the structures are very low. 

Add another figure to overview the health benefits of chlorophyll compounds.

There are still some other activities not reporting in this manuscript, such as anti-inflammatory, neuroprotective etc. 

Add another figure to overview for part 2.2.

Reference to L290

Comments on the Quality of English Language

Edit L104, L283

Author Response

Add reference to Table 1

Response: Thank you for your valuable suggestion, we have added some references in Table 1.

The resolution for the structures are very low. 

Response: Thank you for your comments and suggestions, we have adjusted the figure 1 (as shown above) and figure 5 that can meet the journal's standards.

Figure 5. The molecular structures of SCC salts and their biological functions.

Add another figure to overview the health benefits of chlorophyll compounds.

Response: Thank you for your valuable suggestion, we have added figure 4 to overview the health benefits of chlorophyll compounds.

Figure 4. The health benefits of chlorophyll compounds.

There are still some other activities not reporting in this manuscript, such as anti-inflammatory, neuroprotective etc. 

Response: Thanks to your comments, we have summarized the anti-inflammatory and neuroprotective effects of chlorophyll in lines 365 - 367 and 491 – 493.

Add another figure to overview for part 2.2.

Response: Thank you for your valuable suggestion, we have added figure 3 to overview the function role of chlorophylls.

Fig. 3 The function role of chlorophylls.

Reference to L290

Response: Thank you for your comments and suggestions, we have added reference to line 290.

Edit L104, L283

Response: Thank you for your comments, we have edited line 104 and line 283.

This manuscript is a resubmission of an earlier submission. The following is a list of the peer review reports and author responses from that submission.

Round 1

Reviewer 1 Report

Comments and Suggestions for Authors

The review by Sun et al. is dedicated to multifaceted aspects of structure, biosynthesis, and practical aspects of using chlorophylls from microalgae for human benefit. The review is well-structured, the coverage of literature is fair also. Overall, such a review would constitute certain value for the readership of the journal. However, is still contains some gaps. Fore example, nothing or almost nothing is said about photodynamic cancer therapy with derivatives of chlorophylls and porphyrins. Then, the role of chlorophyll of photosensitizer is not elucidated likewise. Finally, the biotechnological approaches and limitation for maximizing chlorophyll production by microalgal cultures are mentioned only in passing. The text needs a thorough proofreading to get rid of the style and grammar flaws as well as of factual errors (please see the list of specific comments below).

The title needs revision: the structure of diverse chlorophyll species (and their derivatives) in microalgae is well known, so it might be reasonable to highlight the potential health benefits of these compounds. “Microalgae chlorophyll” is grammatically incorrect, it should be either “chlorophylls of microalgae” or “microalgal chlorophylls”.

L150: Chl a is found also in cyanobacteria, not only in plants in the broad sense of the word.

L156: Light reactions of photosynthesis generate not only the reducing power, but also convert the light energy to ATP.

L162: Chl does not transfer the energy of electronic excitation to Car in the course of photosynthesis, only accepts the energy from the Car. If Car take the excitation energy from Chl, this energy is dissipated into heat but not used for photosynthesis. Car never transfer energy to the RCs directly, only via other Chl.

LL165-168: please (i) give the corresponding references and (ii) mention specific pigments.

LL173-178: please provide reference(s).

 L181: I am not aware of pathways for conversion of cholesterol into Car. Both known pathways of Car de novo biosynthesis do not include cholesterol.

Comments on the Quality of English Language

The text needs a thorough proofreading to get rid of the style and grammar flaws.

Reviewer 2 Report

Comments and Suggestions for Authors

Chlorophyll is a crucial pigment in photosynthesis and is abundantly present in microalgae. Due to their simple structure, microalgae are considered excellent sources for chlorophyll extraction. This manuscript, through in-depth exploration of the structure, function, and extraction methods of, lays the theoretical foundation for the standardization and commercial production of chlorophyll. It reviews the potential applications of chlorophyll in the fields of antioxidant, antibacterial and antitumor, holding significant implications for meeting the growing demand for pigments and expanding the application domains of chlorophyll. After careful review and thoughtful consideration, I regret that the manuscript does not currently meet the publication standards of Marine Drugs. The main reasons for rejecting the manuscript are as follows:

1. Relevance of Topic and Content: The content of the manuscript does not align with the theme of microalgae chlorophyll. The manuscript fails to discuss the differences between microalgae chlorophyll and higher plant chlorophyll, making it less relevant to the context of microalgae. Furthermore, the manuscript contains substantial content about the artificial pigment, sodium copper chlorophyllin (SCC), which is fundamentally distinct from the chlorophyll mentioned in the title.

2. Concept Confusion: The concept of natural chlorophyll and synthetic chlorophyll derivative-SCC are confused. This confusion occurs throughout Chapter4. For instance, in section 4.3, the author directly treats SCC as chlorophyll (lines 385-386). Consequently, the entire section discusses the effects of SCC, yet chlorophyll is used as the title in Figure 4, which is an apparent error. It let me think that the consideration for the manuscript is not very rigorous.

3. Structural and Logical Issues: The content of the article appears disorganized, lacking a clear purpose or significance. Also, the arrangement of structure diagram lacks logical order, appearing randomly placed in Figure 1.

Comments on the Quality of English Language

1. Several lengthy sentences could be broken down for clarity. For example, Line 19.

2. Correct the tense issues in certain sentences of the article.

3. Eliminate unnecessary or incorrect usage of words such as "with a view to" (line 23), "of" (line 46), "with a view" (line 82), and "to" (line 216).

4. Clarify expressions that may be confusing, like "culture medium" (line 42).